# Serological Investigation of Bovine Toxoplasmosis Using Commercial and Indigenous ELISA Kits While Validating Cattle Toxo IgG ELISA Kit

**DOI:** 10.3390/ani12162067

**Published:** 2022-08-14

**Authors:** Haroon Akbar, Muhammad Zubair Shabbir, Ubaid Ullah, Muhammad Imran Rashid

**Affiliations:** 1Department of Parasitology, University of Veterinary and Animal Sciences, Lahore 54000, Pakistan; 2Institute of Microbiology, University of Veterinary and Animal Sciences, Lahore 54000, Pakistan

**Keywords:** cattle toxoplasmosis, IgG ELISA, LAT, seropositive, seronegative, sensitivity, specificity

## Abstract

**Simple Summary:**

Toxoplasmosis is a zoonotic disease caused by *T. gondii* infection. The main aims of this study were to assess the seropositivity to toxoplasmosis of an exotic breed of cattle (n = 400) from different farms using the Latex Agglutination Test, validate the Cattle Toxo IgG ELISA kit with the help of the commercially available ELISA kit and compare the efficacy of the LAT and Cattle Toxo IgG ELISA kit for assessing seropositivity of cattle to toxoplasmosis. Four hundred cattle sera were collected from an exotic breed of cattle in Pakistan. These sera were evaluated through an LAT and Cattle Toxo IgG ELISA kit. Of 400 samples, 90 were elected and screened through a commercially available ELISA kit. The sensitivity and specificity of the Cattle Toxo IgG ELISA kit came out to be 100% and 96.15%, and in LAT, it was found as 26.31% and 61.53%, respectively. The Cattle Toxo IgG ELISA kit revealed 29.75% (119/400) seropositivity, 6% less than that found through LAT. The results of this study show that Cattle Toxo IgG ELISA kit is a highly accurate and reliable serodiagnostic tool to diagnose bovine toxoplasmosis.

**Abstract:**

*Toxoplasma gondii* is a protozoan parasite that causes toxoplasmosis in warm-blooded vertebrates, globally. The main aims of this study were to assess the seropositivity to toxoplasmosis of an exotic breed of cattle (*n* = 400) from different farms using the Latex Agglutination Test and validate Cattle Toxo IgG ELISA kit. Of a total of 400 cattle sera that were evaluated by LAT, 143 (35.75%) were found positive. Based on these data, 90 samples (*n* = 60 seronegative by LAT; *n* = 30 seropositive by LAT) were elected for screening through a commercially available ELISA kit. The same 90 samples were screened through a Cattle Toxo IgG ELISA kit for validation purposes. Of 90 samples, 40 were seropositive in the Cattle Toxo IgG ELISA kit (100% sensitivity), and 38 were seropositive in a commercially available ELISA kit. All 50 samples in the Cattle Toxo IgG ELISA kit (96.15% specificity) were also seronegative in the commercially available ELISA kit. Hence, the sensitivity and specificity of the Cattle Toxo IgG ELISA kit came out to be 100% and 96.15%, and in LAT, it was found as 26.31% and 61.53%, respectively. Therefore, the Cattle Toxo IgG ELISA kit is a highly reliable serodiagnostic tool to diagnose bovine toxoplasmosis.

## 1. Introduction

Toxoplasmosis is an important zoonosis caused by a heteroxenous obligate intracellular protozoan named *Toxoplasma gondii* [1]. It causes toxoplasmosis in warm-blooded vertebrates including mammals, rodents, and birds worldwide [2]. Cats, including all *Felidae*, are its definitive hosts and excrete environmentally resistant oocysts in their feces [3]. These oocysts after sporulation can keep their infectivity for several months in soil and water and become responsible for infection of the definitive host and intermediate hosts, including humans and other animals [4]. Humans and other vertebrate intermediate hosts, including sheep, goats, cattle, and buffaloes, acquire toxoplasmosis when they ingest sporulated oocysts or food or water contaminated with sporulated oocysts, whereas meat-eating intermediate hosts, including man, dog, and cat, can also acquire toxoplasmosis after ingestion of raw or undercooked meat of infected animals [5,6,7]. Toxoplasmosis can cause severe problems in cattle, sheep, and goats through fetus absorption, stillbirth, and abortion, consequently leading to economic loss [8]. Cattle have a high natural resistance to *T. gondii,* and it causes subclinical infection in cattle [9]. The economic losses mainly due to toxoplasmosis are in the form of reduction in milk production, stillbirth, and complication related to post abortion such as delay in fertilization and vaginal infection [10]. The seroprevalence of toxoplasmosis in Northern India, Bombay (India), Bangladesh, and Afghanistan was found in cattle at 19.3%, 64.44%, 12%, and 15.74%, respectively [11]. In Pakistan, the overall seroprevalence of toxoplasmosis is higher in humans (65% to 71%) than rats (58.57%), goats (52%), dogs (28.43%), cats (26.43%), cattle (25%), and sheep (24%) [11].

Atypical strains of *Toxoplasma gondii* (UVAS-Toxo-1, UVAS-Toxo-3, UVAS-Toxo-6) have also been reported from Pakistan while studying SAG2 sequence on feline fecal samples [12] such as South America, Central America, the Caribbean, and Africa. [13]. The *T. gondii* has genetic diversity due to genomic recombination, distinct population structures, and intercontinental and regional diversity [14].

Serological surveillance is a good method for identifying and evaluating farm infection [15,16,17]. The serological techniques are relatively accurate, inexpensive, and require a small volume of samples, and these are used in live animals [17,18,19]. Different techniques such as the Sabin–Feldman Dye Test **(**DT), Indirect Fluorescent Antibody Test (iFAT), Modified Agglutination Test (MAT), and Enzyme-linked Immunosorbent Assay (ELISA) are used for the detection of antibodies against *T. gondii* in animals. Subsequently, the OIE Manual declares the DT as the gold standard for detecting antibodies against *T. gondii* in humans [20,21]. It is a time-consuming serological method, requires live virulent Toxoplasma tachyzoites, and has proven unreliable in some species. Since live Toxoplasma is required to do this test, it carries the potential risk for human infection. [21]. In addition, this technique is unavailable in Animal Disease Diagnostic Laboratories in different districts of Punjab Province, Pakistan [22]. In LAT, the soluble antigen is coated on the latex particles that lead to the appearance of beads after reaction with antibodies in positive sera, whereas no bead develops in negative sera. LAT is rapid and easy to perform to detect the anti-*T. gondii* antibodies [23]. It is considered a reliable reference and initial screening test, whereas the DT seems expensive and time-consuming, and its interpretation is difficult related to the operator concern [24,25]. iFAT is an appropriate test for evaluating serum samples collected from live mammals and can be exploited for tissue samples collected during necropsy. The modified agglutination test is suitable for samples with hemolysis because it is more sensitive to serum but has a drawback of false-positive test results [26]. Since the above-said serologic assays have several inherent limitations, the ELISA technique is preferred for screening large quantities of samples [24,27]. Indeed, using purified recombinant proteins, ELISA is routinely used for seroepidemiological investigations with much more reproducibility nowadays [28].

The different recombinant proteins were previously used to detect specific IgG anti-toxoplasma antibodies in animals. These diagnostic antigens are mainly based on surface antigens such as SAG1, SAG2, SAG3 [29,30,31,32,33,34], dense granules such as GRA1, GRA2, GRA4, GRA5, GRA6, GRA7 [30,35,36,37,38], microneme antigens such as MIC2, MIC3, MIC4, MIC5 [23,39], matrix antigens such as MAG1 [40], and rhoptry antigens such as ROP1, and ROP2 [30,41]. The recombinant antigens of *T. gondii* in a single form are used to identify anti-toxoplasma antibodies, whereas the combination form of antigens has been reviewed for increasing their sensitivity [42]. The ELISA is a more reliable and satisfactory technique for identifying antibodies against *T. gondii* compared to the Modified Agglutination Test (MAT) in serology [43]. In the previous studies, SAG1 was used for ELISA in infected mice with the help of the project of Grand Challenges Canada (Grant # S4_0266–01) [44]. The SAG1 is a good candidate for serodiagnosis of toxoplasmosis using cloned genes of *T. gondii* [45,46]. It is an immunodominant antigen highly conserved in nature and used to identify different strains of *T. gondii* [47,48]. Native SAG1 protein has six intramolecular cysteine bridges, making it an immunodominant antigen with immunologically conserved epitopes. The antigens on the surface of an infectious agent carry the highest probability of coming into contact with the reactive antibodies. If the surface antigens are also conserved, it further strengthens their candidacy for utilization in immunodiagnostic assays [45,49]. Subsequently, it can identify acute and chronic toxoplasmosis [50]. The development of the local diagnostic Cattle Toxo IgG ELISA kit was to diagnose bovine toxoplasmosis in Pakistan, and to improve the sensitivity and specificity compared to commercial diagnostic ELISA kits. The main advantages of our local diagnostic ELISA kit are firstly, it will be easily accessible. Secondly, it is economical because a small quantity of recombinant protein of *T. gondii* is required to coat the ELISA plate. This study was designed to assess the seropositivity of the exotic breed of cattle/bovine toxoplasmosis while simultaneously validating our newly developed ELISA kit named as Cattle Toxo IgG ELISA kit exploiting the commercial ELISA kit as Gold Standard.

## 2. Materials and Methods

### 2.1. Collection of Cattle Sera

The present investigation was conducted on the Holstein Friesians breed of cattle (*n* = 400) from four different cattle farms in Pakistan. All the animals were healthy and female belonging to various age groups: 0–1.5 years (*n* = 400), 2–2.5 years (*n* = 76), 3–3.5 years (*n* = 62), 4–4.5 years (*n* = 50) and ≥5 years (*n* = 161).

### 2.2. Screening of Sera by Latex Agglutination Test (LAT)

The LAT was performed to evaluate 400 cattle sera described by protocol [51,52] (Toxo-latex, Linear chemicals, Spanish, latex kit Lot# 20072712). Shortly, each serum sample was diluted (1:4 with 0.1 M PBS). Twenty-five µL of serum was mixed on the blackish region of the glass slide. Positive and negative control sera were mixed by spatula. The slide was rotated by a rocker (3 to 5 min). The agglutination was observed by the naked eye and declared as positive and negative through a comparison of positive and negative controls.

### 2.3. Expression and Purification of rSAG1

The expression and purification of rSAG1 of *T. gondii* was done as described earlier [51]. Briefly, the transformation of pET28a-SAG1 in competent cells of BL21 (DE3) strain was done with heat-shock therapy in the water bath (42 °C for 90 sec.). Immediately after heat shock, the tube was shifted on ice for 30 min. Then, 800 µL of warmed LB broth was added to the mixture and was incubated at 37 °C, 0.66× *g* for one h. The 100 µL of the aliquot was spread on LB agar plates containing kanamycin 1 mg/mL. The plates were incubated at 37 °C for 24 h. After 24 h, restriction analysis confirmed the transformation using restriction enzymes Nhe1 and Xho1. The BL21 (DE3) strain colonies were elected from the LB agar plate (kanamycin 1 mg/mL). The colonies were inoculated in 10 mL of fresh LB broth and incubated (37 °C, 0.66× *g* for 24 h) until the OD_600_ reached 2.0. After attaining the desired concentration, the culture was diluted 100-fold in 50 mL of fresh LB broth, and it was incubated (37 °C, 0.66× *g* for 4 h) until the OD_600_ reached 0.5. After that, 1.0 mM Isopropyl-β-D-thiogalactopyranoside (IPTG) was added to the culture and incubated (37 °C, 0.66× *g* for 6 h). Subsequently, 12% Sodium dodecyl sulfate-polyacrylamide gel electrophoresis (SDS-PAGE) was done. The rSAG1 protein of *T. gondii* was purified by HisPur nickel-nitrilotriacetic acid (Ni-NTA) affinity columns (Thermo Scientific, Pierce Biotechnology, Rockford, Illinois, IL, USA, Cat # 88228).

The rSAG1 was quantified by Bicinchoninic Acid (BCA) assay kit (G-Biosciences, Saint Louis, MO, USA, Cat # 786-570). The purified rSAG1 was verified through WB by immunoblotting with Anti-6x His Tag Antibodies (ThermoFisher, Invitrogen, Houston, TX, USA, Cat # MA1-21315).

### 2.4. Screening of Sera by Cattle Toxo IgG ELISA Kit

The 400 cattle sera were evaluated through the Cattle Toxo IgG ELISA kit described in the protocol [51]. The rSAG1 was coated on a 96-well flat bottom, polystyrene plate (JET BioFil, Hong Kong, China, Code # TCP011096) at 0.125 µg/mL in coating buffer (50 mM Na_2_CO_3_), and it was incubated at 4 °C overnight. The plate was washed with the washing buffer (0.001 M PBS/0.05% Tween-20) at 300 µL/well. The plate was saturated with 4% BSA in 0.01 M PBS at 200 µL/well, and incubated at 37 °C for 2 h. The plate was rewashed five times with the washing buffer. Four hundred cattle serum samples were screened through Cattle Toxo IgG ELISA kit. The positive and negative control sera were dispensed in duplicate wells. Two wells were put as blanks in the plate. The plate was incubated at 37 °C for 2 h. After rewashing five times, Anti-bovine IgG (Whole molecule)- Alkaline Phosphatase, antibodies produced in rabbits (Sigma-Aldrich, Saint Louis, MO, USA, Cat # A0705) were dispensed with 1:10,000 dilution at 100 µL/well and the plate was incubated at 37 °C for 2 h. After rewashing five times, the substrate p-nitrophenyl phosphate (pNPP) (Thermo Scientific, Pierce Biotechnology, Rockford, Illinois, IL, USA, Ref # 34045) was added with 1 mg/mL of Diethanolamine (DEA) substrate buffer (Thermo Fisher Scientific, Loughborough, Leicestershire, UK, Cat # 34064) at 100 µL/well, and incubated at 37 °C. The reaction was stopped after 15 min by adding a stop solution (1 M NaOH) at 100 µL/well. The OD was taken at 405 nm by a Microplate ELISA reader (ELX-800, Tennessee BioTek, Winooski, VT, USA).

### 2.5. Screening of Sera by ID Screen Toxoplasmosis Indirect Multi-Species (Standard ELISA Kit)

Among the 400 sera, 90 were elected for screening through the commercial ELISA kit. Of these, 10 were positive with Cattle Toxo IgG ELISA kit and LAT. An additional 30 were negative for Cattle Toxo IgG ELISA kit, another 20 were positive for LAT. Another 30 were negative through the Cattle Toxo IgG ELISA kit and LAT. The detail of the election is shown in Table 1.

ID Screen Toxoplasmosis Indirect Multi-species (standard ELISA kit) (ID.vet innovative Diagnostics, Louis Pasteur, Grabels, France, Cat # TOXOS-MS-2P) was purchased from ID.vet Company. The standard ELISA was used as per the manufacturer’s instructions. The pre-coated plate with rSAG1 was added with diluted sera (1:10). The unbound antibodies were washed away by the washing solution. Conjugate attached with HRP was added in wells for detecting primary antibodies in the test sera. The antibody-antigen complex was revealed by adding Tetramethylbenzidine (TMB) solution, and the reaction was stopped by adding stop solution. The results were compared with controls. Serum samples, and negative and positive controls were added in duplicate with 1:10 dilution. A well was used as a blank. The plate with sera was incubated at 26 °C for 30 min, and washed three times with a washing solution. Then, 100 µL of enzyme conjugate was added to each well. The plate was incubated at 26 °C for 30 min, and washed again three times with a washing solution. The 100 µL TMB substrate was added in each well. The plate was incubated at 26 °C for 15 min. Then, 100 µL of stop solution (1 N HCl) was added to each well. After 15 min of addition of substrate, the reading of OD was taken at 450 nm by a Microplate ELISA reader (ELX-800, Tennessee BioTek, Winooski, VT, USA).

### 2.6. Difference in Cattle Toxo IgG ELISA Kit and Commercial ELISA Kit

The sensitivity of our kit came out to be 100% against 82.48% of the Standard kit. The optimum serum dilution for Cattle Toxo IgG ELISA kit is 1:100, whereas in the Standard ELISA kit, it is 1:10. Thus, our kit can be exploited in case of minimal amount of serum sample. Cattle Toxo IgG ELISA kit exploits an AP-conjugate, whereas the Standard ELISA kit exploits HRP-conjugate. Cattle Toxo IgG ELISA kit can be optimally incubated at 37 °C , whereas for the Standard ELISA kit, it is 26 °C. Stop solution in our kit is NaOH, whereas it is HCl in the Standard kit.

### 2.7. Statistical Analysis

The MedCalc statistical software (version 11.4.4.0) was used to compare the performance of the Cattle Toxo IgG ELISA kit, LAT, and the standard ELISA kit by calculating the kappa value. ROC curve was used to determine the cut-off value of the Cattle Toxo IgG ELISA kit [53]. The seroprevalence was compared by using Chi-square through SPSS [54].

## 3. Results

### 3.1. Performance of LAT and Cattle Toxo IgG ELISA Kit

Four hundred samples were evaluated through a LAT and a Cattle Toxo IgG ELISA kit. Among these, 143 (35.75%) were positive through LAT, whereas 119 (29.75%) were positive through the Cattle Toxo IgG ELISA kit (Table 2).

The diagnostic performance of LAT and the Cattle Toxo IgG ELISA kit was assessed through positive and negative sera, identified by performing LAT and Cattle Toxo IgG ELISA kit. LAT’s sensitivity and specificity percentages relative to Cattle Toxo IgG ELISA kit were 24.36 and 59.43, respectively. The kappa value was calculated through SPSS as –0.153. The sensitivity and specificity of LAT were found too low relative to the Cattle Toxo IgG ELISA kit. The Pearson Chi-square (χ2) value was 9.551 and was significant at *p* < 0.05 as described in Table 3.

### 3.2. Assessment of Sera by ID Screen Toxoplasmosis Indirect Multi-Species (Standard ELISA Kit)

A total of 90 samples were elected for evaluation through commercial ELISA to check the association of a Cattle Toxo IgG ELISA kit and a standard ELISA kit on the criteria of positive and negative sera by LAT and a Cattle Toxo IgG ELISA kit. The 95% sera were observed positive by the standard ELISA kit for 40 samples. In contrast, for 50 sera, the 100% samples were found negative by the standard ELISA kit as shown in Table 4.

### 3.3. Validation of New ELISA Kit by Standard (Commercial) ELISA Kit

#### 3.3.1. Determination of Sensitivity and Specificity

For the validation of the Cattle Toxo IgG ELISA kit, 90 bovine sera were elected. Of these 90 sera, 30 were seropositive and 60 were seronegative as per Latex Agglutination Test. The elected (90) sera were checked/screened through commercial ELISA kit (ID Screen Toxoplasmosis Indirect Multi-species ELISA kit) that found 38 samples as seropositive and 52 as seronegative. These findings of the commercial standard ELISA kit were taken from a gold standard kit. The same 90 sera were then screened through a Cattle Toxo IgG ELISA kit that determined 40 samples as seropositive (including all the 38 found positive in standard ELISA kit), whereas 50 were found as seronegative. With this information, we calculated the sensitivity and the specificity of the Cattle Toxo IgG ELISA kit as described in Table 5, Table 6 and Table 7.

The sensitivity and specificity of Cattle Toxo IgG ELISA kit emerged to be 100% and 96.15%, respectively, as shown in Figure 1 [51].

#### 3.3.2. Determination of Cut-off Value

The OD values obtained by the Cattle Toxo IgG ELISA kit and through the standard ELISA kit were fed in MedCalc software to determine the cut-off value of the Cattle Toxo IgG ELISA kit that came out to be 0.431. This cut-off value for the new ELISA kit was used to determine the positive and negative bovine sera tested through the Cattle Toxo IgG ELISA kit.

#### 3.3.3. Estimation of Reliability (through Kappa Value)

The kappa value, a frequently used statistical test that measures the inter-rater reliability for qualitative items and interprets the chance of agreement, was calculated for the Cattle Toxo IgG ELISA kit through SPSS and came out to be 0.955. It was found to be highly significant (*p* < 0.001) between the Cattle Toxo IgG ELISA kit and the standard ELISA kit as described in Table 8 and Table 9, thus indicating high reliability of the Cattle Toxo IgG ELISA kit.

#### 3.3.4. Determination of Accuracy (through ROC Analysis)

The Receiver Operating Characteristic (ROC) curve is a graphical plot that describes the diagnostic ability of the binary classifier system. The AUC of the Cattle Toxo IgG ELISA kit was 0.997. The value of the Area Under Curve (AUC) was shown to be highly significant (*p* < 0.001) through ROC. The ROC analysis depicted the importance of the cut-off value of the new ELISA kit. Similarly, the AUC explains the overall performance of diagnostic test as described earlier [51]. The highest accuracy is seen when its value is more than 0.9. The AUC of the new ELISA kit (0.997 > 0.9) indicates that this kit is highly accurate for diagnosing positive and negative sera of toxoplasmosis in cattle.

### 3.4. Serological Investigation of Antibodies against T. gondii in the Exotic Breed of Cattle

Using a Cattle Toxo IgG ELISA kit, seroprevalence of bovine toxoplasmosis in the exotic breed of cattle at different farms was detected as 29.75% (119/400) (*p* < 0.001).

The highest seroprevalence (10.5%; 42/400) was seen in the age group of ≥5 years, followed by 7.5% (30/400) in the age group of 2–2.5 years, 5.5% (22/400) in the age group of 3–3.5 years, 3.75% (15/400) in the age group 4–4.5 years and 2.5% (10/400) in the age group 0–1.5 years (*p* < 0.0001) (See Table 10).

Using LAT, the seroprevalence of bovine toxoplasmosis in the exotic breed of cattle at different farms was detected as 35.75% (143/400) (*p* < 0.001).

The highest seroprevalence (12%;48/400) was seen in the age group of ≥5 years, followed by 09% (36/400) in the age group of 2–2.5 years, 06% (24/400) in the age group of 3–3.5 years, 05% (20/400) in the age group 4–4.5 years and 3.75% (10/400) in the age group 0–1.5 years (*p* < 0.0001), respectively as described in Table 10.

## 4. Discussion

Serological techniques are used to diagnose *T. gondii* infection in animals [23] because tissue cysts of *T. gondii* cannot be detected antemortem. Therefore, detecting antibodies in cattle sera is the exclusive tool for antemortem diagnosis of toxoplasmosis. In our findings, 35.75% of cattle were positive for *T. gondii* antibodies by using LAT. Other researchers have already used it to detect *T. gondii* antibodies and found relatively low seropositivity of cattle to toxoplasmosis [57,58] compared to ours. Compared to other animals for seropositivity of toxoplasmosis, the low prevalence of *T. gondii* in cattle could be attributed to the genetic resistance of cattle to *T. gondii* and a sound management system that lessens the contact of cattle with cat or cat wastes [57].

It has been admitted that Toxoplasma infection in animals is acquired through the ingestion of infective oocysts from the environment [59,60] as an opportunity for exposure to *T. gondii* is routinely available. While the animal ages, its cumulative Likelihood of exposure increases. That is why the age of animals is considered one of the most critical factors in prevalence-related studies of Toxoplasmosis in animals [61].

In the current study, a Cattle Toxo IgG ELISA kit was developed with rSAG1 of *T. gondii* using the same methodology as earlier [51] but with minor changes for adapting it to bovines. The coating concentration of rSAG1 for the new ELISA kit was 0.125 µg/mL, and use of the cattle sera was optimized to a dilution of 1:100. Subsequently, the incubation time for sera and antibodies was optimized to be 2 h at 37 °C. After adding substrate, the optimized reading time for OD was found as 15 min. These optimized conditions showed good diagnostic test values for the Cattle Toxo IgG ELISA kit, and the same conditions were then employed for screening 400 cattle samples.

The Standard ELISA kit used in this study was used as a yardstick and is a reliable serological method for detecting IgG antibodies of *T. gondii* in different animals such as sheep, goats, cattle, buffaloes, dogs, and cats. Its sensitivity and specificity are 82.48% and 97.8%, respectively [27]. In a study by Ademola et al., (2013) they used the same ELISA kit (ID Vet Innovative Diagnostic, Montpellier, France) to test the sera samples of the cattle and pigs [3]. The standard kit has a protocol of only 90 min.

LAT, used as an initial screening and reference test for diagnosis of toxoplasmosis, has 79.1% sensitivity and 86.89% specificity as claimed by the manufacturer. In contrast, in our study, the sensitivity and specificity of the Cattle Toxo IgG ELISA kit surpassed these values of LAT, whereas false positivity of LAT has already been reported [51], highlighting the value of the Cattle Toxo IgG ELISA kit. LAT is not an accurate and reliable method for identifying *T. gondii* infection [62]. It was expressed that LAT is not a corroborative test for identifying *T. gondii* infection [63]. IgM antibodies in the serum may react non-specifically with Toxoplasma whole antigen used in LAT, thus causing false positive reactions as has been seen in some other host species during seroprevalence study on toxoplasmosis. False positive reactivity of IgM with Toxoplasma tachyzoites has been speculated earlier. Thus, non-specific interaction of IgM with Toxoplasma and albumin (particularly BSA) and that of whole serum with latex [64]—most probably owing to a mimicry between conserved PAMPs across species—are the possible reasons behind LAT’s false positive reactivity and warrant further investigation. Another important factor that can potentially drive this apparent difference in seroprevalence between LAT (35.75%) and Cattle Toxo IgG ELISA kit (29.75%) might be the fact that both IgM as well as IgG antibodies can interact with the Toxoplasma antigen in the LAT. In contrast, only the IgG antibody is detectable in the Cattle Toxo IgG ELISA kit. Hence, we can speculate that the higher percentage (6% higher in the current study) of animals detected in LAT than in the Cattle Toxo IgG ELISA kit may represent the ratio of animals currently passing through primary exposure with the *Toxoplasma gondii*. Still another fact that can lie behind this difference in seroprevalence observed between these two tests is that the Cattle Toxo IgG ELISA kit carries only one antigen of *T. gondii* against the total antigenic extract exploited in LAT able to interact with a multitude of antibodies. The 400 bovine sera were also evaluated through the Cattle Toxo IgG ELISA kit, and 29.75% of cattle were found positive using the Cattle Toxo IgG ELISA kit. The highest seropositivity rate for *T. gondii*-specific IgG was found in cattle aged ≥5 years.

In this study, no connection was found between health status and seropositivity of toxoplasmosis in sheep. All the seropositive animals displayed satisfactory health status and looked normal. Mostly cattle are susceptible to infection of *T. gondii* but resistant to the development of clinical signs, hence to the disease. Toxoplasma has been identified as one of the opportunistic pathogens in immunocompromised patients. Cattle have strong immunity and they appear healthy [65].

## 5. Conclusions

This study concluded that Cattle Toxo IgG ELISA kit is a highly accurate and reliable serodiagnostic tool as the sensitivity and specificity of Cattle Toxo IgG ELISA kit were found as 100% and 96.15%, respectively. This study concluded that the Cattle Toxo IgG ELISA kit is a more accurate and reliable immunodiagnostic test than LAT for diagnosing bovine toxoplasmosis in cattle.

## Figures and Tables

**Figure 1 animals-12-02067-f001:**
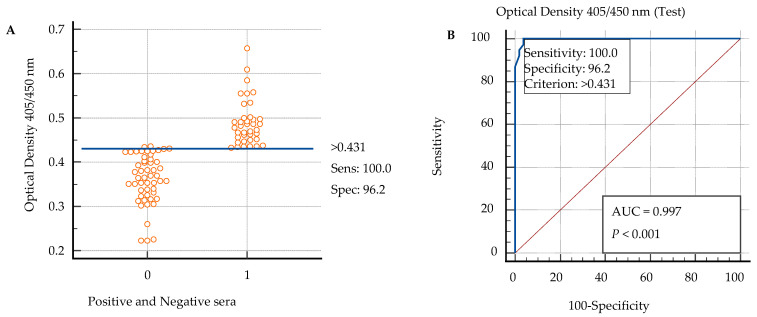
Distribution of toxoplasma-positive and -negative sera detected through Cattle Toxo IgG ELISA kit based on cut-off value (0.431) determined through plotting an ROC curve to find sensitivity and specificity. Sensitivity (Sens), specificity (Spec) (**A**). ROC curve for Cattle Toxo IgG ELISA kit developed through plotting positive versus negative sera identified by Standard ELISA kit to determine the cut-off value of Cattle Toxo IgG ELISA kit (**B**).

**Table 1 animals-12-02067-t001:** Election of sera for standard ELISA kit using LAT, Cattle Toxo IgG ELISA Kit.

Cattle Toxo IgG ELISA Kit and LAT	Samples Elected for Standard ELISA Kit
Cattle Toxo IgG ELISA kit^Positive^ LAT^Positive^	10
Cattle Toxo IgG ELISA kit^Positive^ LAT ^Negative^	30
Cattle Toxo IgG ELISA kit ^Negative^ LAT ^Negative^	30
Cattle Toxo IgG ELISA kit ^Negative^ LAT ^Positive^	20
Total	90

**Table 2 animals-12-02067-t002:** Description of results from LAT and Cattle Toxo IgG ELISA kit for detection of anti-*T. gondii* IgG in 400 sera of cattle.

Cattle Sera	LAT					Cattle Toxo IgG ELISA Kit				
	Results	Seroprevalence	95%CI		*p*-Value	Results	Seroprevalence	95%CI		*p*-Value
			Lower Limit (%)	Upper Limit (%)				Lower Limit (%)	Upper Limit (%)	
Positive	143	35.75%	31.07	40.43	0.001	119	29.75%	25.28	34.22	0.001
Negative	257					281				
Total	400					400				

**Table 3 animals-12-02067-t003:** Performance of LAT relative to Cattle Toxo IgG ELISA kit for detection of anti-*T. gondii* IgG (*n* = 400).

Latex Kit 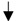	Cattle Toxo IgG ELISA Kit	Total
Positive	Negative
Positive	29	114	143
Negative	90	167	257
Total	119	281	400
Sensitivity = 24.36
Specificity = 59.43
PPV = 20.27
NPV = 64.98
Validity = 41.89
Kappa value = −0.153
Standard error = 0.046
Accuracy = 49
χ2 = 9.551
*p* = 0.002

Positive predictive value (PPV), Negative predictive value (NPV).

**Table 4 animals-12-02067-t004:** The results of 90 cattle sera tested by Standard ELISA kit for detection of anti-*T. gondii* IgG.

Results	Elected Sera	Screening through Cattle Toxo IgG ELISA Kit	Screening through Standard ELISA Kit
Cattle Toxo IgG ELISA kit ^Positive^ LAT^Positive^	10	10	10
Cattle Toxo IgG ELISA kit ^Positive^ LAT ^Negative^	30	30	28
Cattle Toxo IgG ELISA kit ^Negative^ LAT ^Negative^	30	30	30
Cattle Toxo IgG ELISA kit ^Negative^ LAT ^Positive^	20	20	20
Total	90	90	88

**Table 5 animals-12-02067-t005:** Calculation of sensitivity and specificity of Latex Agglutination kit against Gold Standard ELISA kit (Commercial ELISA kit) (*n* = 90).

Latex Kit 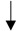	Gold Standard ELISA Kit	Total
Positive	Negative
Positive	10	20	30
Negative	28	32	60
Total	38	52	90
Sensitivity = 26.31%
Specificity = 61.53%

**Table 6 animals-12-02067-t006:** Calculation of sensitivity and specificity of Cattle Toxo IgG ELISA kit against Gold Standard ELISA kit (Commercial ELISA kit) (*n* = 90).

Cattle Toxo IgG ELISA Kit 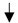	Gold Standard ELISA Kit	Total
Positive	Negative
Positive	38	02	40
Negative	0	50	50
Total	38	52	90
Sensitivity = 100%
Specificity = 96%

**Table 7 animals-12-02067-t007:** Calculation of sensitivity and specificity of Latex Agglutination kit against Cattle Toxo IgG ELISA kit (*n* = 90).

Latex Kit 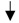	Cattle Toxo IgG ELISA Kit	Total
Positive	Negative
Positive	10	20	30
Negative	30	30	60
Total	40	50	90
Sensitivity = 25%
Specificity = 60%

(Based on the formulae and tables of Thrusfield 2018) [55].

**Table 8 animals-12-02067-t008:** Performance of Cattle Toxo IgG ELISA kit relative to Standard ELISA kit for the detection of anti-*T. gondii* IgG (*n* = 90).

Parameters	Values
Sensitivity	100
Specificity	96.15
PPV	95
NPV	100
validity	99.075
Kappa value	0.955
Standard error	0.032
AUC	0.997
Accuracy	97.77
χ2 = 82.212	
*p* < 0.001	

Positive Predictive Value (PPV), Negative Predictive Value (NPV), Area Under Curve (AUC).

**Table 9 animals-12-02067-t009:** Calculation of Likelihood ratio for a positive-test-outcome ratio (LR+), Likelihood ratio for a negative-test-outcome (LR-), and Diagnostic Odd Ratio (DOR) of our new Cattle Toxo IgG ELISA kit against Gold Standard ELISA kit (Commercial ELISA kit).

Cattle Toxo IgG ELISA Kit 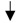	Gold Standard ELISA Test	Total
Positive	Negative
Positive	38	02	40
Negative	0.5	50	50.5
Total	38.5	52	90.5
LR+ = 25.97
LR- = 0.0124
DOR = 2094.35

(Based on the formulae and table of [56]).

**Table 10 animals-12-02067-t010:** Serological investigation of *T. gondii* antibodies in exotic cattle by LAT and Cattle Toxo IgG ELISA kit.

Age Groups (Years)	No. of Examined Animals	LAT				Cattle Toxo IgG ELISA Kit			
		Positive	Sero-Prevalence (%)	Chi-Square (χ²)	*p*-Value	Positive	Sero-Prevalence (%)	Chi-Square (χ²)	*p*-Value
0–1.5	51	15	3.75	χ2 = 32.490	0.0001	10	2.5	χ2 = 65.610	0.0001
2–2.5	76	36	09			30	7.5		
3–3.5	62	24	06			22	5.5		
4–4.5	50	20	05			15	3.75		
≥5	161	48	12			42	10.5		
Total	400	143	35.75			119	29.75		

## Data Availability

The data will be available from corresponding author on request.

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
