# Peer review of "Serological Investigation of Bovine Toxoplasmosis Using Commercial and Indigenous ELISA Kits While Validating Cattle Toxo IgG ELISA Kit"

_animals, 2022, doi:10.3390/ani12162067_

Round 1

Reviewer 1 Report

General comment

The paper describes a new ELISA kit to asses T. gondii seroprevalence in cattle. This new kit, which presented a good performance, was tested against an established latex agglutination test and a comercial ELISA kit was used as gold standard. The new kit was also used to assess seropositivity of 400 exotic breed cattle. Overall this research brings a potential significant improvement in serological analises of cattle with for large scale application.

Specific comments

1)      Suggestion for the authors: after the explanation of the term, use preferentially “Cattle Toxo IgG ELISA kit” or a shorter version of this rather then “our newly developed ELISA kit” or “our new ELISA kit” designations

2)      Previously, we have already identified T. gondii from faecal samples of cats, and these faecal samples were collected from cats at Pet Centre of UVAS (Lahore, Pakistan), which was based on SAG2 sequence, it was atypical strain having UVAS-Toxo-1, UVAS-Toxo-3, UVAS-Toxo-6 of T. gondii in Pakistan [15] whereas, the atypical strain of T. gondii is also found in other countries like in South America, Central America, North America, Carib-bean, and Africa [16].” – the meaning of this sentence is unclear.

3)       “are responsible for infestation of definitive host” – infestation refers to an ectoparasite, as T. gondii is an endoparasite the correct term is infection

4)      The economic losses mainly due to toxoplasmosis are in the form of reduction in milk production, still birth, complication related to post abortion like delay in fertilization, vag-inal infection [10].” – the reference refers to a work in rodents, which has no relation to the effects of toxoplasmosis in cattle or its economic consequences.  This statement should be referenced appropriately.

5)      The prevalence rate of T. gondii antibodies in cattle was 44.8% at herd level whereas, infection rate of herd level were 50% and 33.3% in Khartoum and Gazira States, Sudan, respectively [11].” This sentence resulted from joining 2 sentences from the cited paper’s abstract, however the meaning became unclear. Additionally, there is no rationale for presenting data from Sudan, particularly when very little data is presented on Pakistan, the country focused in the study, and no data is presented relative to Pakistan’s neighboring countries or the Asian continent. This is true for the introduction and the discussion sections. This should be improved/clarified.

6)      DT, IFAT, MAT, and ELISA” – abbreviations should be explained at the first reference, this is an example where this is not the case.

7)      “Subsequently, the OIE Manual declares the Dye Test (DT) as the gold standard for detection of anti-toxoplasma antibodies [23].” This is true only for humans. Also, the paper referenced does not mention toxoplasmosis. The sentence should be clarified and the reference should be corrected, for example by the following “OIE Terrestrial Manual 2018, Chapter 3.10.8. Toxoplasmosis” (https://www.woah.org/fileadmin/Home/eng/Health_standards/tahm/A_summry.htm).

8)       “DAE” is not defined

9)      The titles of Table 3. “Performance of Our new ELISA kit and LAT for the detection of anti-T. gondii IgG in 400 cattle sera” and Table 6. (a) is “Performance of Our new ELISA kit and Standard ELISA kit for the detection of anti-T. gondii IgG in 90 sera” are unclear as the results are relative to the assessment of performance of “Our new ELISA kit” in comparison to LAT. This should be clarified.

10)   The following paragraph must be corrected as the values mentioned in the text from Table 7 correspond to absolute values and not percentages as is written. This also changes the conclusions about the highest rates of seropositivity by age. All this has to be fully reviewed and corrected in the results, discussion and conclusion sections.“In exotic breed, T. gondii antibodies were positive in less than 1.5 years age of cattle through LAT and our new ELISA kit, 15%, and 10%, T. gondii antibodies were positive in 2 to 2.5 years age of cattle through LAT and our new ELISA kit, 36%, and 30%, T. gondii antibodies were positive in 3 to 3.5 years age of cattle through LAT and our new ELISA kit, 24%, and 22%, T. gondii antibodies were positive in 4 or 4.5 years age of cattle through LAT and our new ELISA kit, 20%, and 15%, T. gondii antibodies were positive in more or equal to 5 years age of cattle through LAT and our new ELISA kit, 48% and 42 %, respec-tively as described in Table 7.”

11)   “The sen-sitivity and specificity of our new ELISA kit surpassed these values of LAT whereas false positivity of LAT has already been reported [61], highlighting the value of our new ELISA kit.” This paper also showed apparent false-positives with the LAT, which supports the authors’ previously reported data. This should be discussed in a bit more detail.

12)   It is unclear why the last paragraph of the discussion is a conclusion when there is a conclusions section. This should be corrected.

13)   49 and 61 are the same reference

Author Response

Reviewer 1 Comments :

The paper describes a new ELISA kit to assess T. gondii seroprevalence in cattle. This new kit, which presented a good performance, was tested against an established latex agglutination test and a commercial ELISA kit was used as gold standard. The new kit was also used to assess seropositivity of 400 exotic breed cattle. Overall, this research brings a potential significant improvement in serological analyzes of cattle with for large scale application.

Specific comments

  • Reviewer 1 comments: Suggestion for the authors: after the explanation of the term, use preferentially “Cattle Toxo IgG ELISA kit” or a shorter version of this rather than “our newly developed ELISA kit” or “our new ELISA kit” designations

Response: “Our newly developed ELISA kit” or “our new ELISA kit” have been replaced with “Cattle Toxo IgG ELISA kit” as suggested, throughout the manuscript.

  • Reviewer 1 comments: “Previously, we have already identified  gondii from faecal samples of cats, and these faecal samples were collected from cats at Pet Centre of UVAS (Lahore, Pakistan), which was based on SAG2 sequence, it was atypical strain having UVAS-Toxo-1, UVAS-Toxo-3, UVAS-Toxo-6 of T. gondii in Pakistan [15] whereas, the atypical strain of T. gondii is also found in other countries like in South America, Central America, North America, Carib-bean, and Africa [16].” – the meaning of this sentence is unclear.

Response:

This sentence has been clarified as follows: “Atypical strains of Toxoplasma gondii (UVAS-Toxo-1, UVAS-Toxo-3, UVAS-Toxo-6) have also been reported from Pakistan while studying SAG2 sequence on feline faecal samples like South America, Central America, Caribbean, and Africa.”

(Page no.2, line 74-80)

  • Reviewer 1 comments: “are responsible for infestation of definitive host” – infestation refers to an ectoparasite, as gondii is an endoparasite the correct term is infection.

The term “infestation” has been replaced with “infection” as suggested.

(Page no.2, line  52)

  • Reviewer 1 comments: “The economic losses mainly due to toxoplasmosis are in the form of reduction in milk production, still birth, complication related to post abortion like delay in fertilization, vaginal infection [10].” – the reference refers to a work in rodents, which has no relation to the effects of toxoplasmosis in cattle or its economic consequences.  This statement should be referenced appropriately.
  • Response: A reference related to cattle has been added in the sentence as per your kind suggestions.
  • (Page no.2, line 61-62)
  • Reviewer 1 comments:“The prevalence rate of  gondii antibodies in cattle was 44.8% at herd level whereas, infection rate of herd level was 50% and 33.3% in Khartoum and Gazira States, Sudan, respectively [11].” This sentence resulted from joining 2 sentences from the cited paper’s abstract, however the meaning became unclear. Additionally, there is no rationale for presenting data from Sudan, particularly when very little data is presented on Pakistan, the country focused on the study, and no data is presented relative to Pakistan’s neighboring countries or the Asian continent. This is true for the introduction and the discussion sections. This should be improved/clarified.

Response:

This has been improved/clarified as per your kind suggestions.

(Page no.2, line  62-70)

  • Reviewer 1 comments: “DT, IFAT, MAT, and ELISA” – abbreviations should be explained at the first reference, this is an example where this is not the case.

Response:

This has been explained as per your kind suggestions.

(Page no.2, line  99-102, 104-109)

  • Reviewer 1 comments: “Subsequently, the OIE Manual declares the Dye Test (DT) as the gold standard for detection of anti-toxoplasma antibodies [23].” This is true only for humans. Also, the paper referenced does not mention toxoplasmosis. The sentence should be clarified and the reference should be corrected, for example by the following “OIE Terrestrial Manual 2018, Chapter 3.10.8. Toxoplasmosis” (https://www.woah.org/fileadmin/Home/eng/Health_standards/tahm/A_summry.htm).

Response:

The sentence has been clarified as suggested:

“OIE Manual declares the DT as the gold standard for detecting antibodies against T. gondii in humans. It is time consuming serological method, requires live virulent Toxoplasma tachyzoites and has proven unreliable in some species. Since live Toxoplasma is required to do this test, it carries the potential risk for human infection. In addition, this technique is unreliable in Animal Disease Diagnostic Laboratories in different districts of Punjab province, Pakistan.”

(Page no.2, line  90-96)

The reference has also been corrected as follows.

“OIE, S. Manual of Diagnostic Tests and Vaccines for Terrestrial Animals 2013., Chapter 3.10.18, Toxoplasmosis”

(Page no.16, line 562)

  • Reviewer 1 comments: “DAE” is not defined

Response:

It has been corrected as “DEA” which is “Diethanolamine”.

(Page no.05, line 203)

  • Reviewer 1 comments: The titles of Table 3.“Performance of Our new ELISA kit and LAT for the detection of anti- gondii IgG in 400 cattle sera” and Table 6. (a) is “Performance of Our new ELISA kit and Standard ELISA kit for the detection of anti-T. gondii IgG in 90 sera” are unclear as the results are relative to the assessment of performance of “Our new ELISA kit” in comparison to LAT. This should be clarified.

Response:

The titles have been clarified as follows:

“Table 3: Performance of LAT relative to Cattle Toxo IgG ELISA kit for the detection of anti-T. gondii IgG (n = 400).”

(Page no.07, line 269, 270)

“Table 6 (a): Performance of Cattle Toxo IgG ELISA kit relative to Standard ELISA kit for the detection of anti-T. gondii IgG (n = 90).”

(Page no.10, line 345, 346)

  • Reviewer 1 comments: The following paragraph must be corrected as the values mentioned in the text from Table 7 correspond to absolute values and not percentages as is written. This also changes the conclusions about the highest rates of seropositivity by age. All this has to be fully reviewed and corrected in the results, discussion, and conclusion sections. “In exotic breed,  gondii antibodies were positive in less than 1.5 years age of cattle through LAT and our new ELISA kit, 15%, and 10%, T. gondii antibodies were positive in 2 to 2.5 years age of cattle through LAT and our new ELISA kit, 36%, and 30%, T. gondii antibodies were positive in 3 to 3.5 years age of cattle through LAT and our new ELISA kit, 24%, and 22%, T. gondii antibodies were positive in 4 or 4.5 years age of cattle through LAT and our new ELISA kit, 20%, and 15%, T. gondii antibodies were positive in more or equal to 5 years age of cattle through LAT and our new ELISA kit, 48% and 42 %, respectively as described in Table 7.”

Response:

Table 7 has been revised in the manuscript, now having both absolute values as well as percentages in separate columns.

The description of the table has been further clarified as per your kind suggestions.

(Page no. 12, line 388-390)

  • Reviewer 1 comments: “The sensitivity and specificity of our new ELISA kit surpassed these values of LAT whereas false positivity of LAT has already been reported [61], highlighting the value of our new ELISA kit.” This paper also showed apparent false-positives with the LAT, which supports the authors’ previously reported data. This should be discussed in a bit more detail.

Response:

As suggested, this has been discussed a bit more as follows and added in the manuscript:

“Another important factor that can potentially drive this apparent difference in seroprevalence between LAT (35.75%) and Cattle Toxo IgG ELISA kit (29.75%), might be the fact that both IgM as well as IgG antibodies can interact with the Toxoplasma antigen in the LAT. In contrast, only the IgG antibody is detectable in the Cattle Toxo IgG ELISA kit. Hence, we can speculate that the higher percentage (6% higher in the current study) of animals detected in LAT than in the Cattle Toxo IgG ELISA kit,  may represent the ratio of animals currently passing through primary exposure with the Toxoplasma gondii. Still another fact that can lie behind this difference in seroprevalence observed between these two tests, is that the Cattle Toxo IgG ELISA kit carries only one antigen of T. gondii against the total antigenic extract exploited in LAT- able to interact with a multitude of antibodies.”

(Page no. 14, line 453-463)

  • Reviewer 1 comments: It is unclear why the last paragraph of the discussion is a conclusion when there is a conclusions section. This should be corrected.

Response:

Corrected as per your kind suggestions.

(Page no.14, line 468-473)

  • Reviewer 1 comments: 49 and 61 are the same reference

Response:

The reference 61 has been deleted whereas it has been updated with reference 49 as identified, and now its number in the reference list is 53.

(Page no.17, line 634-635)

Author Response

Reviewer 2 Comments :

Introduction

1). Reviewer 2 comments: The reference 28 is very old. It is better a new reference.

Response:

The reference 28 has been replaced with a new reference as per your kind suggestion.

New Reference: “Reynoso-Palomar et al. (2020) Prevalence of Toxoplasma gondii parasite in captive Mexican jaguars determined by recombinant surface antigens (SAG1) and dense granular antigens (GRA1 and GRA7) in ELISA-based serodiagnosis”

 (Page no.16, line 592-594)

2). Reviewer 2 comments: The text is written about the test IFAT, but it is not explained as LAT, ELISA, and MAT. It is necessary to write something about it.

Response:

“In LAT, the soluble antigen is coated on the latex particles that lead to the appearance of beads after reaction with antibodies in positive sera whereas no bead develops in negative sera. LAT is rapid and easy to perform to detect the anti-T. gondii IgG antibodies.

IFAT is an appropriate test for the evaluation of serum samples collected from live mammals and can be exploited for tissue samples collected during necropsy. For samples with hemolysis, the modified agglutination test is suitable because it is more sensitive to serum but has a drawback of false-positive test results.

iFAT is an appropriate test for evaluating of serum samples collected from live mammals and can be exploited for tissue samples collected during necropsy. The modified agglutination test is suitable samples with hemolysis because it is more sensitive to serum but has a drawback of false-positive test results.

Since the above-said serologic assays have several inherent limitations, the ELISA technique is preferred for  screening  large quantities of samples. Indeed, using purified recombinant proteins, ELISA is routinely used for seroepidemiological investigations with much more reproducibility nowadays.”

(Page no.2, line  99-102, 104-109, 109-113)

“Liu, Q., et al. (2015) Diagnosis of toxoplasmosis and typing of Toxoplasma gondii

“Miller, M., et al. (2002) Evaluation of an indirect fluorescent antibody test (IFAT) for demonstration of antibodies to Toxoplasma gondii in the sea otter (Enhydra lutris)”

“Teimouri, A., (2019) Development, optimization, and validation of an in-house Dot-ELISA rapid test based on SAG1 and GRA7 proteins for serological detection of Toxoplasma gondii infections”

3). Reviewer 2 comments: The reference 45 the text did not explain why the SAG1 was a good candidate to purpose of serodiagnosis. It is necessary to explain.

Response: It has been explained as suggested:

“Native SAG1 protein has six intramolecular cysteine bridges, making it an immunodominant antigen with immunologically conserved epitopes. The antigens on the surface of an infectious agent, carry the highest probability of coming into contact with the reactive antibodies. If the surface antigens are also conserved, it further strengthens their candidacy for utilization in immunodiagnostic assays.”

(Page no.03, line 131-136)

“Cesbron-Delauw, et. al. (1994) Similarities between the primary structures of two distinct major surface proteins of Toxoplasma gondii

2.Materials and Methods

4). Reviewer 2 comments: 2.1 It is necessary to write to age of the animals, the kind of portion is given to animals.

Response: Age groups information has been added in materials and methods as per your kind suggestion.

(Page no.04, line 151-152)

5). Reviewer 2 comments: 2.2. The technique the standard kit ELISA is not described.

Response: The technique has now been described in detail as per your kind suggestion. (Page no.05, line 218-234)

6). Reviewer 2 comments: 2.3. The technique new ELISA is described, but what is the difference with the commercial ELlSA? It is necessary to write in paper.

Response:  The differences have been added in the paper as suggested:

“The sensitivity of our kit came out to be 100% against 82.48% of the Standard kit. The optimum serum dilution for Cattle Toxo IgG ELISA kit is 1:100 whereas in the Standard ELISA kit. It is 1:10. Thus, our kit can be exploited in case of minimal of serum sample. Cattle Toxo IgG ELISA kit exploits an AP-conjugate whereas the Standard ELISA kit exploits HRP-conjugate. Cattle Toxo IgG ELISA kit can be optimally incubated at 37ºC whereas for the Standard ELISA kit, it is 26ºC. Stop solution in our kit is NaOH whereas it is HCl in the Standard kit.”

(Page no. 05 AND 06, line 237-243)

7). Reviewer 2 comments: 2.4. (Standard ELISA kit)

Table 01 is uncompleted and it is not well described. It is necessary to explain better the

numbers of the animals

Response:

The description of table 1 has been improved while explaining the number of the animals as per your kind suggestions.

(Page no.05, line 209-217)

  1. Results

8). Reviewer 2 comments: The tables 5a, 5b e 5c - there is necessary to build a single table. It is better to analyze the results.

Response: Three tables have been merged into one table as per your kind suggestion.

(Page no.08 and 09, line 294-297)

4.Discussion

8). Reviewer 2 comments: The text - The present study was multipurpose: one was to evaluate the serological investigation of bovine toxoplasmosis in exotic breed of cattle at different farms of Pakistan by using Latex Agglutination Test (LAT), second was to validate our new ELISA kit exploiting commercially available ELISA kit as a Gold Standard ELISA Kit for this purpose. Third was to make use of our validated new ELISA kit to determine seroprevalence of bovine toxoplasmosis and to compare its findings with those from LAT. It is objective. The text could be removed.

Response: As indicated, the text has been removed.

(Page no.13, line 393-398)

9). Reviewer 2 comments: The reference 55 is very old. It is could be removed or changed.

Response: As indicated, the reference 55 has been removed, new reference has been added.

(Page no.16, line 565, 566)

The following reference has been added into the manuscript.

“Liu, Q., et al. (2015) Diagnosis of toxoplasmosis and typing of Toxoplasma gondii

10). Reviewer 2 comments: The text - our study, we also calculated the sensitivity and specificity of Latex kit using our newly developed ELISA kit, it appeared to be 26.31% sensitivity and 61.53% specificity. The sensitivity and specificity of our new ELISA kit surpassed these values of LAT whereas false positivity of LAT has already been reported. It is result. It is better to explain the false positivity in LAT and to do the discussion about the sensitivity and specificity.

Response:

“The false-positivity of LAT has been reported elsewhere. They described that this false-positivity might be due to the high concentration of hepatitis B virus "e" antigen as this antigen can give false positive reactions by using LAT while studying the seroprevalence of T. gondii. Another reason might be the cross reactivity of BSA used during the preparation of latex particles. Even this reactivity was not affected through the treatment of sera with 2-mercaptoethanol.”

(Page no.14, line 448-453)

“Jalallou, et. al. (2010) Recombinant SAG1 antigen to detect Toxoplasma gondii specific immunoglobulin G in human sera by ELISA test”

The text - Out of a total of 400 sera samples screened through LAT, 90 were selected for their screening through standard ELISA kit. The same 90 sera were exploited for validation purpose: to determine sensitivity, specificity, cut-off value, kappa value, AUC through ROC analysis, for our new ELISA kit. it is a method. Is it already explain in methodology? It is not necessary. 

Response: This has been deleted from the discussion.

(Page no.13, line 426-429).

The text - For our new ELISA kit development, the cut-off value (calculated using MedCalc software) came out to be 0.431, AUC = 0.997, a highly significant value (p < 0.001), portraying its high accuracy. It is a methodology. Is necessary to explain the accuracy.

Response: It has been deleted from the discussion.

(Page no.13, line 436-438)

In the present study, the correlation was found between health status and serological investigation of toxoplasmosis in cattle. Most of the animals found seropositive, had satisfactory health status, and looked normal, apparently. Why??? It is not discussed.

Response:

“Mostly cattle are susceptible to infection of T. gondii but resistant to the development of clinical signs- hence to the disease. Toxoplasma has been identified as one of the opportunistic pathogens in immunocompromised patients. Cattle have strong immunity and they appear healthy.”

 (Page no.14 line 470-473)

“Esteban-Redondo, et. al., (1997) Toxoplasma gondii infection in sheep and cattle. Comparative immunology, microbiology, and infectious diseases 20, 191-196”

Conclusions

The results of this study concluded that our newly developed ELISA kit is highly accurate and reliable serodiagnostic tool as the sensitivity and specificity of our newly developed ELISA kit found as 100% and 96.15%, respectively. Overall, 29.75% (119/400) cattle sera were found  positive for toxoplasma specific IgG in newly validated ELISA kit whereas the LAT revealed 35.75% (143/400) of these animals as having  toxoplasma specific IgG. Subsequently, 10.5% (42/400) of the cattle in the age-group “5 years or above” were found seropositive while screening through newly developed ELISA kit whereas it was even higher in case of LAT i.e. 12%(48/400). The text present is the result. Is necessary to tell the conclusion.

Response: So, it is not necessary and now it was deleted from the conclusion.

(Page no.14, line 487-496).

“This study concluded that Cattle Toxo IgG ELISA kit is a highly accurate and reliable serodiagnostic tool as the sensitivity and specificity of Cattle Toxo IgG ELISA kit were found as 100% and 96.15%, respectively. This study concluded that the Cattle Toxo IgG ELISA kit is more accurate and reliable immunodiagnostic test than LAT for the diagnosing of bovine toxoplasmosis in cattle.” (Page no.14, line 483-487)

Reviewer 3 Report

Authors have thoroughly evaluated the potential of newly establish ELISA kit as field diagnostic test to detect toxoplasmosis. 

General comments:

Try to avoid starting a sentence with a numerical number.

Section 2.2: Please beifely explain the test procedure. Please briefly explain the methods used in expression and purification of recombinant SAG. 

Table 1: I suggest to change "Our new ELISA kit" to its actual name :Cattle Toxo IgG ELISA kit" in the table. The title of the 2nd column need to change. Isn't it the number of samples elected for standard ELISA kit?

I also suggest if possible, change that terminology in the other parts of the manuscript too after introducing it once in the text.

Section 2.4: Can you please explain the differences, advantages and disadvantages between standard ELISA kit and your Cattle toxo IgD ELISA kit? Either in the introduction or in the discussion.  This is to highlight the superiority of your newly developed ELISA to already available ELISA test. 

Section 2.4: If you have followed the manufacture's instructions please mention that in the 1st sentence of the 2nd paragraph. 

Tables 5a, b and c: There are blue arrows inside these tables. 

Figure 1: This figure appears dragged horizontally, please provide a one without disproportional enlargement. 

Manuscript needs extensive editing of English grammar and writing style. At the moment it is not in a publishable level. 

Author Response

Reviewer 3 Comments :

Authors have thoroughly evaluated the potential of newly establish ELISA kit as field diagnostic test to detect toxoplasmosis. 

 General comments:

1). Reviewer 3 comments: Try to avoid starting a sentence with a numerical number.

Response: Modified as suggested.

(Page no.06, line 254-257, Page no.07, line 277, Page no.01, line 17 )

2). Reviewer 3 comments: Section 2.2: Please briefly explain the test procedure. Please briefly explain the methods used in expression and purification of recombinant SAG. 

Response:

Test procedure has been added briefly as follows:

“Shortly, each serum sample was diluted (1:4 with 0.1 M PBS). Twenty five µl of serum was mixed on the blackish region of the glass slide. Positive and negative control sera were mixed by spatula. The slide was rotated by a rocker (3 to 5 min). The agglutination was observed by the naked eye and declared as positive and negative through a comparison of positive and negative controls.”

 (Page no.04, line 159-163).

Expression and purification of rSAG1 has been added as follows:

“2.3. Expression and purification of rSAG1

The expression and purification of rSAG1 of T. gondii was done as described earlier. Briefly, the transformation of pET28a-SAG1 in competent cells of BL21 (DE3) strain was done with heat-shock therapy in the water bath (42°C for 90 sec.). Immediately after heat shock, the tube was shifted on ice for 30 min. Then, 800µl of warmed LB broth was added to the mixture and was incubated at 37°C, 0.66 x g for one h. The 100 µl of the aliquot was spread on LB agar plates containing kanamycin 1 mg/ml. The plates were incubated at 37°C for 24 h. After 24 h., restriction analysis confirmed the transformation by using restriction enzymes Nhe1 and Xho1. The BL21 (DE3) strain colonies were selected from the LB agar plate (kanamycin 1 mg/ml). The colonies were inoculated in 10 ml of fresh LB broth and incubated (37°C, 0.66 x g for 24 h) until the OD600 reached 2.0. After attaining the desired concentration, the culture was diluted 100 folds in 50ml of fresh LB broth, and it was incubated (37°C, 0.66 x g for 4 h) until the OD600 reached 0.5. After that, 1.0 mM Isopropyl-β-D-thiogalactopyranoside (IPTG) was added to the culture and incubated (37°C, 0.66 x g for 6 h). Subsequently, 12% Sodium dodecyl sulfate-polyacrylamide gel electrophoresis (SDS-PAGE) was done. The rSAG1 protein of T. gondii was purified by HisPur nickel-nitrilotriacetic acid (Ni-NTA) affinity columns (Thermo Scientific, USA, Cat # 88228).

The rSAG1 was quantified by Bicinchoninic Acid (BCA) assay kit (G-Biosciences, MO, USA, Cat # 786-570). The purified rSAG1 was verified through WB by immunoblotting with Anti-6x His Tag Antibodies (ThermoFisher, USA, Cat # MA1-21315).”

(Page no.04, line 166-185)

3). Reviewer 3 comments:  Table 1: I suggest to change "Our new ELISA kit" to its actual name :Cattle Toxo IgG ELISA kit" in the table. I also suggest, if possible, change that terminology in the other parts of the manuscript too after introducing it once in the text.

 The title of the 2nd column needs to change. Isn't it the number of samples elected for standard ELISA kit?

Response:

The name of kit has been modified as suggested across the manuscript.

Yes, these are the samples elected for standard ELISA kit and the title of 2nd column has been changed as suggested.

(Page no.05, line 217)

4). Reviewer 3 comments:   Section 2.4: Can you please explain the differences, advantages and disadvantages between standard ELISA kit and your Cattle toxo IgG ELISA kit? Either in the introduction or in the discussion.  This is to highlight the superiority of your newly developed ELISA to already available ELISA test. 

Response:

“The development of the local diagnostic Cattle Toxo IgG ELISA kit was to diagnose bovine toxoplasmosis in Pakistan, and to improve the sensitivity and specificity compared to commercial diagnostic ELISA kits. The main advantages of our local diagnostic ELISA kit are; firstly, it will be easily accessible. Secondly, it is economical because a small quantity of recombinant protein of T. gondii is required to coat the ELISA plate.”

 (Page no.03, line 137-142)

“The sensitivity of our kit came out to be 100% against 82.48% of the Standard kit. The optimum serum dilution for Cattle Toxo IgG ELISA kit is 1:100 whereas in the Standard ELISA kit. It is 1:10. Thus, our kit can be exploited in case of minimal of serum sample. Cattle Toxo IgG ELISA kit exploits an AP-conjugate whereas the Standard ELISA kit exploits HRP-conjugate. Cattle Toxo IgG ELISA kit can be optimally incubated at 37ºC whereas for the Standard ELISA kit, it is 26ºC. Stop solution in our kit is NaOH whereas it is HCl in the Standard kit.”

(Page no. 05 and 06, line 237-243)

5). Reviewer 3 comments:   Section 2.4: If you have followed the manufacturer’s instructions, please mention that in the 1st sentence of the 2nd paragraph. 

Response: Mentioned as indicated. (Page no.05, line 220, 221)

6). Reviewer 3 comments:   Tables 5a, b and c: There are blue arrows inside these tables. 

Response: Modified as indicated (Page no.08 and 09, line 294-297).

 7). Reviewer 3 comments:  Figure 1: This figure appears dragged horizontally, please provide a one without disproportional enlargement. 

Response:

Corrected as per your kind suggestions.

(Page no.10, line 323, 325)

8). Reviewer 3 comments:  Manuscript needs extensive editing of English grammar and writing style. At the moment it is not in a publishable level. 

Response:

The English grammar and writing style of the manuscript has been edited extensively as per your kind suggestions.

Round 2

Reviewer 2 Report

The review is ok, but i think the reference 10 could be change, because hepatitis B is comun in humans.

The false-positivity of LAT has been reported elsewhere. They described that this false-positivity might be due to the high concentration of hepatitis B virus "e" antigen as this antigen can give false positive reactions by using LAT while studying the seroprevalence of T. gondii. Another reason might be the cross reactivity of BSA used during the preparation of latex particles. Even this reactivity was not affected through the treatment of sera with 2-mercaptoethanol.”

(Page no.14, line 448-453)

Author Response

COMMENTS and Responses:

The review is ok.

1). Reviewer 2 Comments :

But I think the reference 10 could be change.

Response:

The reference 10 has been changed with reference 11 as per your kind suggestions.

(Page no.15, line 533-536)

“Stelzer, et al. (2019).Toxoplasma gondii infection and toxoplasmosis in farm animals: Risk factors and economic impact. Food and Waterborne Parasitology 2019, 15, 1-32.”

2). Reviewer 2 comments:

Because hepatitis B is common in humans.

“The false-positivity of LAT has been reported elsewhere. They described that this false-positivity might be due to the high concentration of hepatitis B virus "e" antigen as this antigen can give false positive reactions by using LAT while studying the seroprevalence of T. gondii. Another reason might be the cross reactivity of BSA used during the preparation of latex particles. Even this reactivity was not affected through the treatment of sera with 2-mercaptoethanol.”

(Page no.14, line 448-453)

Response:

We think that the study cited for discussion, was conducted on Toxoplasma seroprevalence. In addition, the study was conducted in a mammalian species hence we deem it good for discussion. Keeping in view the opinion of our esteemed reviewer, we have modified the text in a context that is now more relevant to our study, and can be seen in the following paragraph. We hope that it would now represent a more clarified context for citing this study and would be acceptable by the reviewer.

“IgM antibodies in the serum may react non-specifically with Toxoplasma whole antigen used in LAT thus causing false positive reactions as has been seen in some other host species during seroprevalence study on toxoplasmosis. False positive reactivity of IgM with Toxoplasma tachyzoites has been speculated earlier. Thus, non-specific interaction of IgM with Toxoplasma and albumin (particularly BSA) and that of whole serum with latex (1) -most probably owing to a mimicry between conserved PAMPs across species, are the possible reasons behind LAT’s false positive reactivity and warrant further investigation.”

(Page no.14, line 448-455)

“Holliman, et al. (1989) Discrepant toxoplasma latex agglutination test results. Journal of clinical pathology 42:200-203.”